# Engineering the Active Site Lid Dynamics to Improve the Catalytic Efficiency of Yeast Cytosine Deaminase

**DOI:** 10.3390/ijms24076592

**Published:** 2023-04-01

**Authors:** Hanzhong Deng, Mingming Qin, Zhijun Liu, Ying Yang, Yefei Wang, Lishan Yao

**Affiliations:** 1Qingdao New Energy Shandong Laboratory, Qingdao Institute of Bioenergy and Bioprocess Technology, Chinese Academy of Sciences, Qingdao 266101, China; 2Shandong Energy Institute, Qingdao 266101, China; 3University of Chinese Academy of Sciences, Beijing 100049, China; 4National Facility for Protein Science, Shanghai Advanced Research Institute, Chinese Academy of Sciences, Shanghai 201210, China

**Keywords:** dynamics engineering, cytosine deaminase, prodrug, protein structure

## Abstract

Conformational dynamics is important for enzyme catalysis. However, engineering dynamics to achieve a higher catalytic efficiency is still challenging. In this work, we develop a new strategy to improve the activity of yeast cytosine deaminase (yCD) by engineering its conformational dynamics. Specifically, we increase the dynamics of the yCD C-terminal helix, an active site lid that controls the product release. The C-terminal is extended by a dynamical single α-helix (SAH), which improves the product release rate by up to ~8-fold, and the overall catalytic rate *k*_cat_ by up to ~2-fold. It is also shown that the *k*_cat_ increase is due to the favorable activation entropy change. The NMR H/D exchange data indicate that the conformational dynamics of the transition state analog complex increases as the helix is extended, elucidating the origin of the enhanced catalytic entropy. This study highlights a novel dynamics engineering strategy that can accelerate the overall catalysis through the entropy-driven mechanism.

## 1. Introduction

Enzymes can accelerate chemical reactions by orders of magnitude. The great catalytic power arises not only from the enzyme structure but also from the conformational dynamics. The static structure provides covalent and noncovalent (such as electrostatics, h-bonding, and hydrophobic stacking) interactions, stabilizing the transition state. The dynamics create an ensemble of conformations that help to guide the reactant through the catalytic process, i.e., substrate binding, formation of a transition state, and product release. The importance of enzyme dynamics in catalysis has been well documented [1,2,3,4,5,6,7,8,9,10,11,12,13,14]. In some circumstances, the conformational dynamics was found to be the limiting factor for the overall catalytic rate [15]. Therefore, it is a challenging and interesting question whether one could enhance the catalytic efficiency of an enzyme by manipulating its conformational dynamics [16,17,18,19].

Recently, an engineering study targeting dynamics was carried out for an ancestral luciferase, where the transplantation of a dynamic structural fragment at the entrance of the active site significantly enhances the substrate binding kinetics and thus the overall catalytic rate, indicating the potential of dynamics engineering [20]. In addition, the modification of substrate binding tunnel dynamics has also been adopted to change the selectivity of an aminomutase [21]. Moreover, earlier studies also underline the usefulness of dynamics engineering in directed evolution [22,23]. However, questions remain whether the enzyme catalysis is improved through the modulation of conformational dynamics, which is directly related to conformational entropy [24]. If the dynamics engineering perturbs the enzyme structure, the catalysis might be improved through an enthalpic effect although the dynamics is targeted.

In this work, we developed a new engineering strategy that enhances the dynamics, entropy, and overall catalysis of yeast cytosine deaminase (yCD), which can convert 5-fluorocytosine (5-FC) to 5-fluorouracil (5-FU), a chemotherapeutic agent that is widely used in cancer therapy. The yCD/5-FC system is of great biomedical interest in the application of enzyme prodrug therapy [25,26,27,28], which can minimize the deleterious side effects on healthy cells of patients undergoing chemotherapy. The improved enzyme would provide additional benefits by decreasing the number of cells that have to express or target yCD, and thus decreasing the bystander effect in which 5-FU can diffuse into and kill neighboring healthy cells. A few attempts have been made to improve the yCD activity [28,29]. Using 5-FC as the substrate, a triple mutant A23L/V108I/I140L improves *k*_cat_ by 57%, whereas a single mutant D92E improves *k*_cat_ by 69%. However, combing the two mutants decreases *k*_cat_ [29]. Site-saturation mutagenesis was performed to screen all single mutants of yCD [26]. A few single mutants, including C71A, E75P, and H127V, were identified and combined. The activity was quantified by the IC_50_ of 5-FC on HT1080 cell lines expressing yCD variants. The best mutant C71A/H127V shows an IC_50_ close to the A23L/V108I/I140L mutant [26], suggesting that it is challenging to improve yCD catalytic efficiency significantly by mutating its own sequence. Recently, the A23L/V108I/I140L mutant of yCD has been used in Phase III clinical trials for recurrent high-grade glioma and Phase I trials for metastatic solid tumors [30,31].

The yCD catalytic mechanism has been well studied [32,33,34,35]. In the chemical deamination step, E64 acts as a proton shuttle in facilitating the reaction [32]. The overall catalysis is limited by the product release [33]. The C-terminal helix (P149–E158) acts as a lid to cover the active site [36,37] in the transition state analog (TSA) complex and apo forms. The lid helps sequester the catalytic residues and the substrate from the solvent, but also hinders the product release. In this work, we explored a new strategy to accelerate the yCD catalysis by improving the dynamics of the C-terminal helix. Instead of modifying the amino acid sequence of the C-terminal helix itself, we decided to extend the C-terminal by a dynamical single α-helix (SAH) [38,39,40] (Figure 1A). It is known that protein dynamics is coupled to solvent dynamics [41]. The extension would increase the protein solvent coupling and thus enhance the conformational dynamics as it becomes more exposed to the solvent. Moreover, a more solvent-accessible fragment also increases the solvation entropy, which may benefit the product release as well. Together with directed evolution, NMR, and X-ray crystallography, we show that the yCD product release and overall catalysis can be enhanced through dynamics engineering and that entropy plays an essential role in the improved catalytic activity.

## 2. Results

### 2.1. Construct yCD with a Helix Tail

The C-terminal helix of yCD (P149–E158) covers the active site and limits the product release and the overall catalysis. To increase the product release rate, the dynamics of the C-terminal helix has to be boosted. Highly-charged single α-helix (SAH) is a group of sequences which can form a stable α-helical conformation. Here, we extended the yCD C-terminal helix by a 68-residue SAH, corresponding to residues 918–985 of *S. scrofa* myosin-VI [38,39,40], with a 5-residue linker added in between (Figure 1B). The inserted linker helps remove the potential steric clashes between yCD and SAH. The sequence optimization of the linker was carried out through directed evolution. Random mutagenesis of the linker was screened by the crude enzyme catalysis using 5-FC as the substrate. The optimal linker sequence RIELQ was identified. The fusion protein (named “yCD-RQ-SAH”) was purified, and its specific activity was measured. The yCD-RQ-SAH has an activity 2.8-fold of the wt-yCD (enzyme concentration 0.05 uM and 5-FC concentration 10 mM) (Figure 1C). To dissect the contribution from different components, the activity of yCD with only the RQ linker (yCD-RQ) was also measured. This variant has an activity 1.8-fold of the wt-yCD. Adding 1/8 of SAH to yCD-RQ (namely yCD-RQ-1/8SAH) increases the activity to 2.5-fold of the WT. Adding 1/4 and 1/2 of SAH to yCD-RQ increases the activity to 2.9- and 3.2-fold of the WT, respectively. The extension of the C-terminal helix gradually increases the enzyme activity until it reaches 1/2 SAH, which has the maximum activity (Figure 1C).

To gain further insight into the activity improvement mechanism, more detailed kinetic studies were performed for the catalysis of 5-FC. Michaelis−Menten kinetic constants *k*_cat_ and *K*_m_ were determined through the nonlinear fitting of reaction rate versus the 5-FC concentration (Figure 2A). The gradual *k*_cat_ increase (until yCD-RQ-1/2SAH) matches well with the overall enzyme-specific activity increase (Figure 1B and Figure 2B). Meanwhile, *K*_m_ also generally increases as the C-terminal helix is extended until it reaches yCD-RQ-1/2SAH. For yCD-RQ and yCD-RQ-1/8SAH, the increase in *K*_m_ is small so that the enzyme specificity is improved due to *k*_cat_ increase. For yCD-RQ-1/4SAH, yCD-RQ-1/2SAH, and yCD-RQ-SAH, the *K*_m_ increase is larger than that of *k*_cat_. As a result, these three variants have a smaller enzyme specificity compared to the wt-yCD, although they have a higher turnover rate.

### 2.2. Product Release Rate of yCD Variants

As suggested previously [33], product release is the rate-limiting step of the wt-yCD 5-FC catalysis. One might speculate that the increase in *k*_cat_ is caused by a faster product release step. The product dissociation rate was measured using a 1D 19F saturation transfer NMR experiment. There are two tryptophan residues, W10 and W152, in the enzyme [33]. The 19F labeling by 5-fluorotryptophan in yCD gives two well-separated peaks in the spectrum (Figure 3A). The binding of the product 5-FU shifts the W10 signal downfield. A saturation transfer experiment was performed to extract the product dissociation rate k_off_ (Figure 3A). The k_off_s of yCD-RQ and yCD-RQ-1/8SAH are 3.5- and 5.7-fold the WT, respectively, suggesting that the product release becomes more efficient (Figure 3B). As the C-terminal helix is extended, k_off_ becomes much larger than *k*_cat_, suggesting that the product release no longer limits the overall catalysis. For example, yCD-RQ-1/8SAH has a koff of 108 s^−1^, ~ 2-fold of its *k*_cat_; yCD-RQ-1/4SAH has a k_off_ of 175 s^−1^, 3.2-fold of its *k*_cat_ (Figure 2B and Figure 3B).

### 2.3. Activation Enthalpy and Entropy of yCD Variants

To determine the activation enthalpy and entropy of yCD and its variants for the substrate 5-FC catalysis, kinetic measurements were performed at 283, 293, 298, 303, 308, and 313 K. The Eyring equation was used to extract ΔH^‡^ and ΔS^‡^ (Figure 4A). The positive ΔH^‡^ and negative ΔS^‡^ indicate that both enthalpy and entropy tend to increase the activation barrier, unfavorable to the catalysis. As the C-terminal helix extends, ΔH^‡^ increases, whereas TΔS^‡^ increases as well but with a larger amplitude (Figure 4B), suggesting that the decrease in the activation free energy of yCD variants is caused by the increase in activation entropy (less negative). From the enthalpic point of view, the transition state is destabilized, suggesting that some of the transition state interactions might be weakened due to increased entropy, a typical effect of enthalpy–entropy compensation. The kinetic data suggest that the transition state protein dynamics is modified to accelerate the catalysis.

### 2.4. Structure and Active Site Interactions of yCD-RQ and yCD-RQ-1/8SAH

To reveal the structural basis of the activity enhancement of yCD variants, the protein crystal structures of yCD-RQ (PDB ID: 8I3N) and yCD-RQ-1/8SAH (PDB ID: 8I3O) were solved (An attempt to crystalize yCD-RQ-1/4SAH and yCD-RQ-1/2SAH failed for unknown reasons.). yCD-RQ and yCD-RQ-1/8SAH have comparable structural features, with the overall structure presenting no significant deviation from the wt-yCD. However, minor structural variations were identified at the C-terminal helix (Figure 5A). The helix exhibits a ~7° rotation away from the enzyme active site in comparison with the apo wt-yCD (PDB ID: 1ox7), resulting in the binding pocket being more exposed to the solvent environment. Consequently, the active site residues W152 and D155 move away from the original position, which weakens the stacking between the W152 side chain and the aromatic ring of F114, as well as the hydrogen bonds between the carboxylate of D155 and the side chains of N51 and H62 (Figure 5B). The weakened interaction might increase the flexibility of the C-terminal helix. This is supported by a more disordered structure appearing in the C-terminus, since E158 cannot be observed in the crystal structure of yCD-RQ or yCD-RQ-1/8SAH. The electron density of the RQ linker and 1/8SAH was poor for modeling in either the yCD-RQ or yCD-RQ-1/8SAH structure due to high flexibility.

In addition, as the C-terminal helix is extended, the overall catalytic rate k_cat_ becomes smaller than k_off_, the product dissociation rate, suggesting that the transition state of the chemical step plays a more important role in limiting the overall catalysis. Upon examining the interactions between the chemical step transition state and surrounding residues, the structure of yCD-RQ-1/8SAH in the complex with the inhibitor 2-hydroxypyrimidine was solved (PDB ID: 8I3P). The 2-hydroxypyrimidine was converted by yCD into 4-(R)-oxidanyl-3,4-dihydro-1H-pyrimidin-2-one, the TSA [36]. In general, the complex still forms a homodimer with a slight conformational divergence at the C-terminus between the two protomers. The C-terminal helix is restored to a structure similar to the wt-yCD after binding the TSA (Figure 6A). The high-quality electron density map of the linker and part of the 1/8 SAH was clearly observed in this structure, showing that residues 156–161 form a loop structure while residues 162–167 form an α-helical structure, as expected for the SAH. The X-ray B-factor indicates that the RQ linker and the observed 1/8 SAH are very flexible (Figure 6B).

The active site interaction of yCD-RQ-1/8SAH is comparable to the wt-yCD TSA complex, indicating that the TSA binding was overall unaffected by the extension at the C-terminal helix (Figure 6C), in line with the kinetic measurements that the activation enthalpy of yCD-RQ-1/8SAH is only 0.3 kcal/mol higher than the wt-yCD (Figure 4B). The carboxyl group of E64 moves slightly toward the TSA (by 0.1 Å). How this perturbation contributes to the catalysis is hard to predict because the change is so small. The very similar active site interactions of the two proteins suggest that the enhanced catalysis is unlikely from the enthalpy change.

### 2.5. Conformational Dynamics from H/D Exchanges

To characterize the dynamics of yCD in the chemical step transition state, H/D exchange experiments were performed for various yCD TSA complexes (including wt-yCD, yCD-RQ, yCD-RQ-1/8SAH, and yCD-RQ-1/4SAH) using NMR. Six residues in the C-terminal helix display measurable H/D exchange rates in wt-yCD and/or the yCD variants (Figure 7). For example, E154 has the H/D exchange rates of 0.78, 2.5, 3.6, and 7.1 h^−1^ in wt-yCD, yCD-RQ, yCD-RQ-1/8SAH, and yCD-RQ-1/4SAH, respectively; I156 has the corresponding rates of 0.24, 1.5, 2.3, and 3.9 h^−1^. All six residues show higher exchange rates as the C-terminal helix is extended, indicating that the helix dynamics is enhanced by the C-terminal extension (Figure 7). Although the active site structure of yCD-RQ-1/8SAH is almost the same as the wt-yCD (Figure 6C), distinctive dynamics properties are observed between the two proteins. Because the yCD TSA complex is a mimic of the chemical step transition state, it is expected that the conformational entropy of the transition state increases as well when the helix is extended. The H/D exchange data provide direct evidence that yCD dynamics is enhanced to accelerate the catalysis through an entropic mechanism.

## 3. Discussion

The importance of conformational dynamics to enzyme catalysis is well known. However, engineering dynamics to achieve higher catalytic efficiency is challenging. In this work, by targeting the dynamics of the C-terminal helix, an active site lid that controls the product release and seals the active site for catalysis, we successfully improve the catalytic efficiency of yCD. The C-terminal helix is extended by SAH, a stable α-helix, with a five residue linker RQ inserted in between (Figure 1B). The RQ linker was used to accommodate SAH. Although attaching SAH to yCD directly (yCD-SAH, Appendix A) also improves the enzyme activity by 1.3-fold (16% lower than yCD-RQ-SAH), it may cause potential steric clashes. The dynamics of SAH has been studied before [40]. The helix shows a wobbling motion around the long axis, with the amplitude depending on the length of the helix. As the helix becomes longer, the motion amplitude becomes larger. This dynamic property is corroborative with the yCD activity observation that the longer SAH helix tends to accelerate the catalysis more efficiently (Figure 1C). However, it is not clear that increasing the helix length from 1/2 SAH to SAH slightly decreases the enzyme activity. As the turnover rate increases, the *K*_m_ constant also becomes larger as if the substrate binding becomes weaker when the C-terminal helix is extended (Figure 2). The X-ray structure of yCD-RQ in the apo form has a C-terminal helix tilted slightly away from the active site, which may weaken the interaction with the substrate (Figure 5).

It is known that for the wt-yCD, the rate-limiting step is the product release [33] which is hindered by the C-terminal helix motion [42], even though the dissociation constant of the product is rather large, ~25 mM [33]. With the SAH fusion tag being elongated, the product release rate increases significantly (Figure 3) and becomes larger than *k*_cat_ (Figure 2), so that it is no longer the rate-limiting step. It appears that the motion of the C-terminal helix is accelerated by the SAH extension.

The gain of the catalytic efficiency is mainly driven by the rise in the activation entropy (Figure 4A), whereas the increase in activation enthalpy partially offsets the entropic effect (Figure 4B). Since the product release is not the limiting factor of the overall catalysis, it is important to check the transition state of the chemical step. The hydration of the added inhibitor 2-hydroxypyrimidine to 4-(*S*)-hydroxyl-3,4-dihydropyrimidine in yCD crystallization creates a good TSA. The X-ray structure of the yCD-RQ-1/8SAH TSA complex shows that the active site interactions are well preserved when compared with the wt-yCD complex. However, the SAH helix displays high flexibility, as suggested by its large B-factors (Figure 6B). In comparison, the C-terminal helix shows slightly elevated dynamics as well compared to the protein rigid part, but its flexibility is much smaller than that of SAH. It is not clear from B-factors how the SAH dynamics is related to the C-terminal helix dynamics (Figure 6B).

H/D exchange using NMR is a well-established method to study protein stability, dynamics, and solvent accessibility [43]. Except for a few protein amides with the slowest H/D exchanging rate due to global unfolding, most residues exchange through a local fluctuation between the closed and open states [44]. Therefore, the exchanging rate is a good indicator for protein dynamics [45]. In yCD and its variants in the complex with TSA, the C-terminal helix has a relatively fast exchanging rate compared to amides of the core β-sheet residues (Appendix A). The increase in the exchanging rate indicates that the local fluctuation of residues in the C-terminal helix increase as it is extended (Figure 7). This corresponds to an increase in the conformational entropy of the C-terminal helix in the TSA complex, which explains that the reaction catalyzed by yCD variants (compared to the wt-yCD) is facilitated by the increase in the activation entropy (Figure 4). In other words, the extension of the yCD C-terminal helix using the α-helix SAH increases its dynamics, which accelerates the product release and chemical reaction through an entropy-driven mechanism. As the yCD activity increases, it may pose some toxicity to E. *Coli* cells. We did not observer any dramatic change in the yCD protein expression level among different variants. An increase in expression (by less than 1-fold compared to the wt-yCD) was noticed for the 15N-labeled yCD variants with longer helix tails, probably due to better solubility. Since these variants have higher activities, toxicity seems not to be a problem.

## 4. Materials and Methods

### 4.1. Construction of Mutagenesis Library

The plasmid that encodes yCD-GGGGS-SAH with a TEV protease-cleavable N-terminal His-tag was constructed by Personalbio Technology Co., Ltd. (Shanghai, China), and used as the template to construct the mutagenesis library. Five amino acid residues GGGGS were selected to be replaced by random amino acids with the target codon of NNS (N = A/T/C/G in a 1:1:1:1 ratio, S = C/G in a 1:1 ratio) by polymerase chain reaction (PCR). The products were transformed into *E. coli* BL21(DE3) pLysS competent cells and cultivated on LB agar plates with ampicillin and chloramphenicol.

### 4.2. Screening Procedure

The colonies of mutagenesis library were inoculated into a 96-deep-well plate containing 1 mL LB medium with 100 µg/mL ampicillin and 20 µg/mL chloramphenicol. After being cultivated at 37 °C for 12 h, the seed solutions were transferred into another 96-deep-well plate containing 0.5 mL LB medium with the same antibiotics. The first plate was stored at −80 °C after the addition of 30% glycerol. The clones in the second plate were cultivated at 37 °C for 4 h, then induced with 0.5 mM isopropyl *β*-D-thiogalactopyranoside (IPTG) and 1 mM zinc acetate, and further incubated at 16 °C for 12h. *E. coli* cells were collected by centrifugation (4 °C, 3000 g, 20 min), washed twice with the reaction buffer (20 mM Tris, 100 mM NaCl, pH 7.0), resuspended in 3 mg/mL lysozyme, and incubated at 37 °C, 800 rpm for 1 h. The cell lysate was collected by centrifugation and a BCA assay kit was used to quantify the total protein concentration. After incubating 5-FC with the protein for 30 min, the reaction was terminated by 0.1 N HCl. By detecting absorption at the wavelength of 255 nm, a characteristic absorbance frequency of the product 5-FU [46], using a SpectraMax M2*^e^* spectrophotometer (Molecular Devices), the clones of yCD variants with high activity were marked. After DNA sequencing of the same clones in the first plate, the positive mutant sequences were identified. The screening process was carried out for a total of ~5000 colonies.

### 4.3. Expression and Purification of yCD Fusion Proteins

PCR was used to construct the plasmid-encoded yCD fusion proteins with the SAH of different lengths (yCD-RQ, yCD-RQ-1/8SAH, yCD-RQ-1/4SAH, and yCD-RQ-1/2SAH). The expression and purification protocols have been described previously [35]. Here, we carried out the procedure in a slightly different way. For unlabeled yCD proteins, LB medium was used for protein expression, which was induced by the addition of 0.5 mM IPTG and 1 mM zinc acetate when the OD_600_ of the culture reached 0.8. The cells were then incubated at 16 °C overnight under shaking (200 rpm). For isotope-labeled yCD proteins, BL21(DE3) pLysS cells containing desired plasmids were grown in 1 L LB medium first. When the OD_600_ reached 0.8, the cells were centrifuged and resuspended in 500 mL M9 medium. To obtain ^19^F-labeled proteins, 5-fluorotryptophan (a final concentration of 50 mg/L) was added to the M9 medium. After being grown for 30 min, the protein expression was induced. The *E. coli* cells were harvested by centrifugation, washed once with buffer A (20 mM Tris, 1 M NaCl, pH 7.5), and kept at −20 °C for further usage. All the proteins were purified with the Ni-NTA agarose column. After removing the His-tag with TEV protease, the flowthrough was purified further through a Sephadex G-75 column. The purity of proteins was monitored by SDS-PAGE, whereas the concentration of the proteins was determined by OD_280_.

### 4.4. Kinetic Measurements and Data Fitting

The enzymatic activities of wt-yCD and fusion proteins were determined by measuring the conversion rates from 5-FC to 5-FU. To determine the specific activity of wt-yCD and its variants, a mixture (50 μL) of 5-FC (10 mM) and yCD (0.05 μM) was incubated in the reaction buffer (20 mM Tris, 100 mM NaCl, pH 7.0). The reaction was quenched by adding 200 μL 0.1 N HCl after a certain time of reaction (1, 4, 8, 12, and 16 min). The concentration of 5-FU was determined by measuring the absorbance value at the wavelength of 255 nm. The reaction rate was obtained from the linear fitting of 5-FU concentration versus the reaction time.

The same experimental procedure was used to determine the *K*_m_ and *k*_cat_ values for 5-FC deamination by wt-yCD and its variants at different temperatures. The concentration of 5-FC was varied between 0.15 and 19.2 mM. The standard Michaelis−Menten equation was applied to fit the kinetic *K*_m_ and *k*_cat_ values; Δ*S*^‡^ and Δ*H*^‡^ for the catalytic reaction were fitted with the Eyring equation.

### 4.5. ^19^F NMR Spectroscopy and the Product Release Rate

The 1D ^19^F saturation transfer NMR experiment was carried out on a Bruker Avance 400 MHz NMR spectrometer to measure the 5-FU dissociation rate following the previous work [33]. For the wt-yCD, the sample contained 1.5 mM ^19^F tryptophan-labeled protein and 25 mM 5-FU. For the yCD fusion proteins, the same protein concentration was used but with 75mM 5-FU (approximately the maximum soluble concentration) due to weaker binding. To obtain the dissociation rate, the ^19^F NMR signal of the bound form W10 was saturated. Ten spectra were acquired for each saturation transfer experiment with saturation times of 0.05, 24, 36, 48, 72, 96, 120, 144, 240, and 360 ms. The control experiment was performed with the saturation frequency set at one end of the spectrum. All the 1D NMR spectra were processed using TopSpin and the peak volume of the W10 free form was fitted to yield the product release rate [33].

### 4.6. H/D Exchange Rate of yCD and Its Variants

Lyophilized wt-yCD, yCD-RQ, yCD-RQ-1/8SAH, and yCD-RQ-1/4SAH powders were dissolved with 100 μL of 50 mM 2-hydroxypyrimidine hydrochloride inhibitor solution (20 mM Tris, 50 mM NaCl, pH 7.0, 100% H_2_O). Then, 400 μL 99.9% D_2_O was added to the protein TSA complex solution. The 2D ^1^H–^15^N HSQC spectra (10 min each for a total of 6 h) were collected consecutively at 308 K. All the HSQC spectra were processed and analyzed using NMRPipe and NMRDraw [47]. The exchange rates were obtained by fitting the 2D ^1^H–^15^N HSQC peak height of exchanged residues to a two-parameter exponential function. The errors were estimated from duplicate measurements of two prepared samples. The peak assignment for the 2D ^1^H–^15^N HSQC spectrum of yCD-RQ-1/8SAH were conducted with a series of 3D sequential assignment experiments including HNCO, HN(CA)CO, HNCACB, and HN(CO)CACB for a 1mM ^2^H/^13^C/^15^N-labeled yCD-RQ-1/8SAH sample (10 mM inhibitor, 20 mM Tris, 50 mM NaCl, pH 7.0, 95%/5% H_2_O/D_2_O). The ^1^H–^15^N HSQC peaks for other fusion proteins were assigned (up to residue E158) through comparison with that of yCD-RQ-1/8SAH.

### 4.7. Crystallization, Data Collection, and Structure Refinement

yCD mutants were screened for conditions for both apo enzymes and enzymes complexed with TSA at 291 K using commercially available screening kits. Initial crystallization conditions were found by the sitting-drop diffusion method. After optimization and additive screening, conditions were found by the hanging-drop diffusion method to successfully grow large single crystals for later data collection. The apo yCD-RQ was crystallized by mixing 1 μL of protein solution containing 50 mg/mL yCD-RQ and 1 μL of reservoir solution containing 0.15 M ammonium acetate, 27% (*w/v*) PEG3350. The apo yCD-RQ-1/8SAH was crystallized by mixing 1 μL protein solution containing 50 mg/mL yCD-RQ-1/8SAH and 1 μL reservoir solution containing 0.15M ammonium acetate, 29% (*w/v*) PEG3350. The yCD-RQ-1/8SAH complexed with TSA was crystallized by mixing 1 μL protein solution containing 80 mg/mL yCD-RQ-1/8SAH, 25mM 2-hydroxypyrimidine, and 1 μL reservoir solution containing 1.5 M ammonium sulfate, 0.1 M Tris pH 8.5, 12% (*v/v*) glycerol. The crystals were directly transferred into liquid nitrogen and stored for later data collection. The X-ray data of apo yCD-RQ and apo yCD-RQ-1/8SAH were collected at BL02U1 beamline of the SSRF (Shanghai Synchrotron Radiation Facility) with an DECTRIS EIGER2 S 9M detector and the yCD-RQ-1/8SAH complex was collected from BL10U2 beamline with an DECTRIS EIGER X 16M detector under 100K. The dataset was collected for 180 degrees with a 0.5-degree gap. The collected data were processed with Aquarium [48]. Statistics for data collection are summarized in Appendix A.

All the mutant structures were solved by full-feature molecular replacement (MR) in PHENIX [49] using the wild-type yCD structure (PDB ID:1ox7) as the model. The generated model was further manually built by COOT [50] based on the extra electron density and refined by PHENIX-Refine [51]. The overall quality of the finalized structures was assessed by MolProbity [52]. The data statistics for refinement of the structures are also summarized in Appendix A.

PyMOL version 1.7 (https://www.schrodinger.com accessed on 1 March 2022) was used to perform structural analysis of the protein crystal structures and to generate all figures. For apo wt-yCD and the apo yCD-RQ, the monomer was selected for structural alignment in modeling. As for wt-yCD and the yCD-RQ-1/8SAH complexed with TSA, the dimer formed by chain A and chain B was chosen for structural alignment in modeling due to the slight difference between two chains in the C-terminal of yCD-RQ-1/8SAH.

## 5. Patents

A Chinese patent has been filed from the work reported in this manuscript.

## Figures and Tables

**Figure 1 ijms-24-06592-f001:**
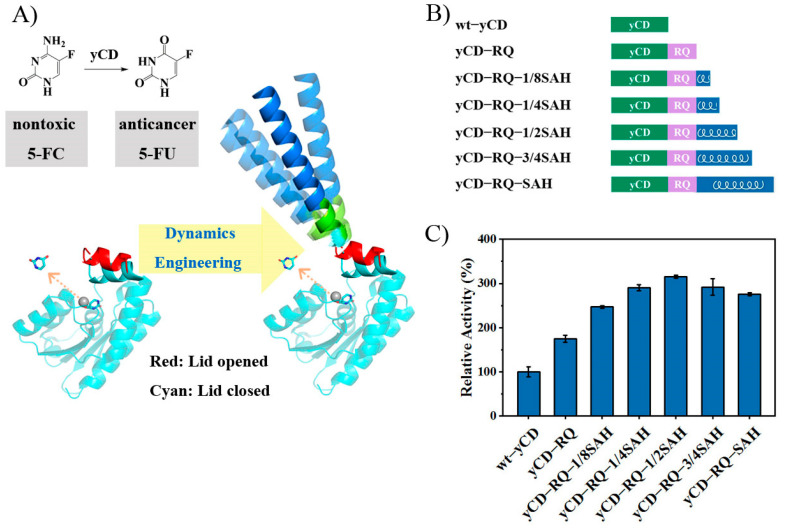
Constructs of yCD variants and their relative activity. (**A**) Schematic drawing of the strategy to accelerate the yCD catalysis (5-FC to 5-FU) by engineering the dynamics of the C-terminal helix (which controls the rate-limiting product release) through the helix extension with a single α-helix SAH. The dynamics of SAH may enhance the dynamics of the C-terminal helix. (**B**) yCD constructs with the SAH helix (or its part) and a linker RQ (composed of five residues) in between. (**C**) Relative catalytic activity against 5-FC, with the wt-yCD activity set to 100%. The reaction mixture contained 10 mM 5-FC and 0.05 μM yCD in a buffer of 20 mM Tris and 100 mM NaCl, pH 7.0. The reactions were carried out at 303 K in triplicate. The error bars are the standard deviation.

**Figure 2 ijms-24-06592-f002:**
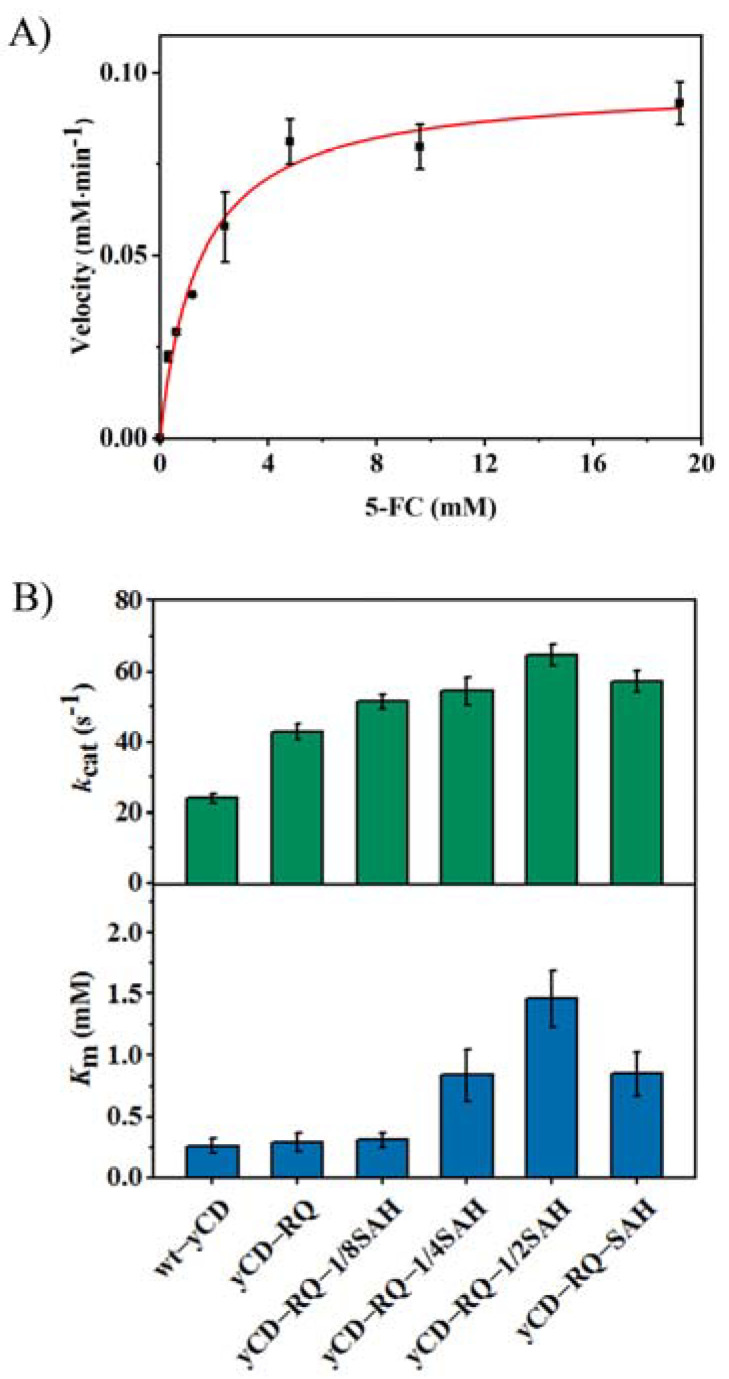
Michaelis−Menten kinetic constants *k*_cat_ and *K*_m_ for wt-yCD and its variants on 5-FC catalysis: (**A**) fitting of the Michaelis−Menten equation for the catalysis by yCD-RQ-1/2SAH; (**B**) *k*_cat_ and *K*_m_ values of different yCDs. The reactions were carried out at 303 K in 20 mM Tris and 100 mM NaCl, pH 7.0. The error bars are the standard deviation from triple experiments.

**Figure 3 ijms-24-06592-f003:**
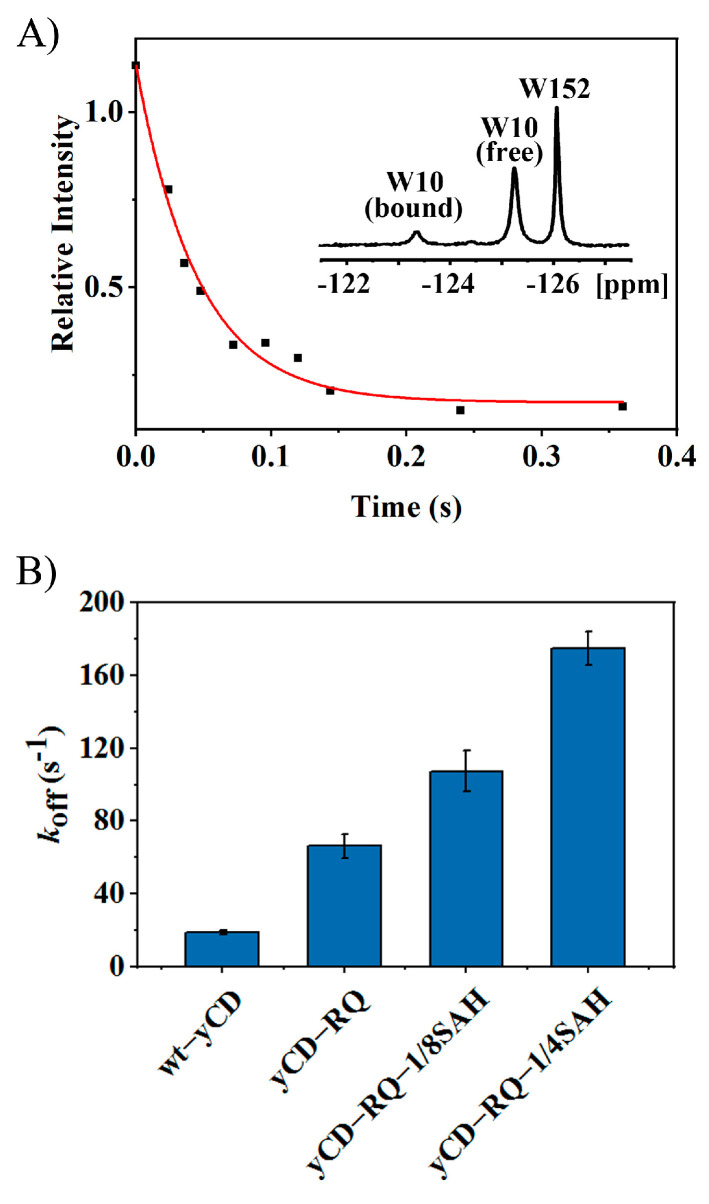
Product dissociation rate of yCD variants obtained from the 1D ^19^F saturation transfer NMR experiment. The binding of 5-FU shifts the W10 signal downfield. (**A**) Signal intensity decay of the W10 free form after the saturation of the bound form signal of yCD-RQ-1/8SAH. Inserted is the 1D ^19^F spectrum of 5-fluorotryptophan-labeled yCD-RQ-1/8SAH (1.5 mM) in the presence of the product 5-FU (75 mM). (**B**) The dissociation rate k_off_ obtained from the nonlinear fitting of the signal decay curve. The error bars are from the goodness of fitting.

**Figure 4 ijms-24-06592-f004:**
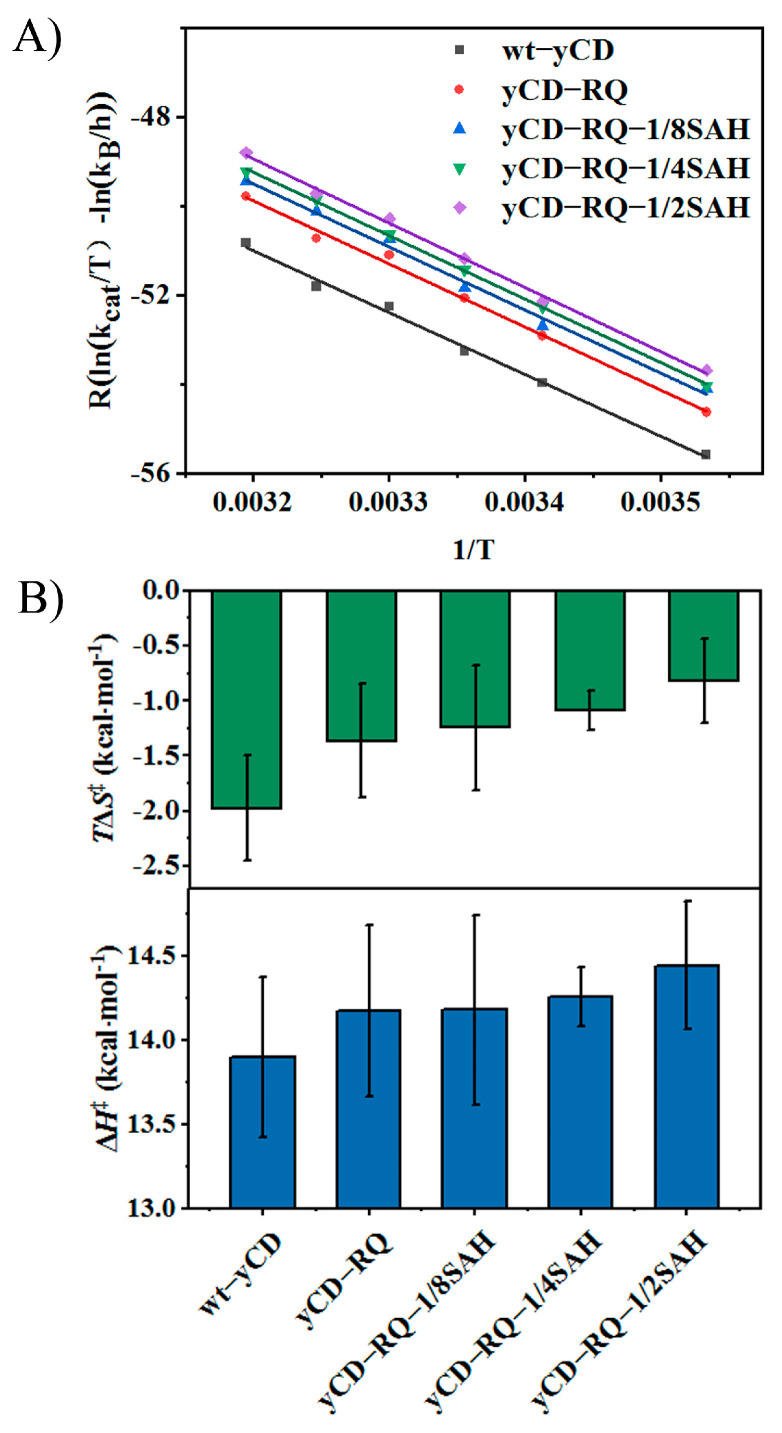
Activation enthalpy ΔH^‡^ and entropy ΔS^‡^ from the Eyring equation: (**A**) Eyring plot of the catalytic rate constant *k*_cat_ for the substrate 5-FC; (**B**) ΔH^‡^ and TΔS^‡^ obtained from the linear fitting. Both ΔH^‡^ and TΔS^‡^ increase as the C-terminal helix is extended, suggesting that the higher catalytic activity of yCD fusion proteins is driven by increased activation entropy. The error bars in panel (**A**) are standard deviation from triplicate experiments. The error bars in panel (**B**) are from the goodness of linear fitting in panel (**A**).

**Figure 5 ijms-24-06592-f005:**
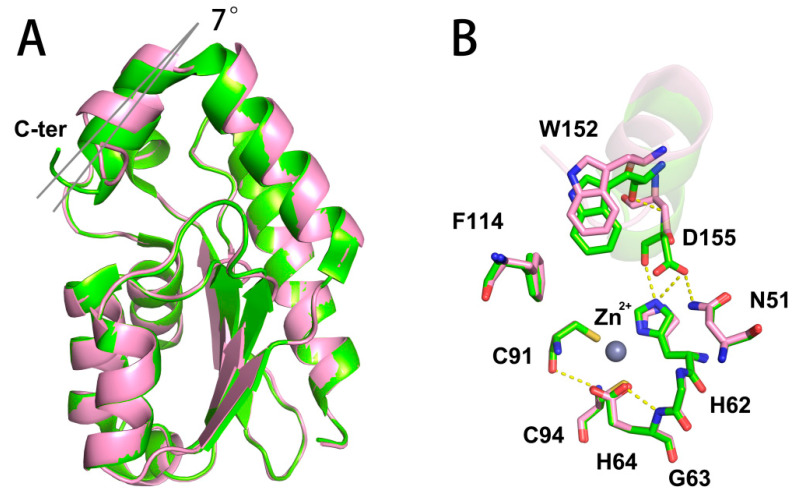
Structural comparison between yCD-RQ and wt-yCD protomers. (**A**) The overall view of yCD-RQ (pink) and wt-yCD (green). The C-terminal helix lid (labeled as C-ter) of yCD-RQ slightly tilts away from the active site. (**B**) Zoomed-in view of the active sites. The active site residues are shown in sticks and the C-terminal helix is shown in cartoon. The hydrogen bonds in the active site are labeled in yellow dashed lines.

**Figure 6 ijms-24-06592-f006:**
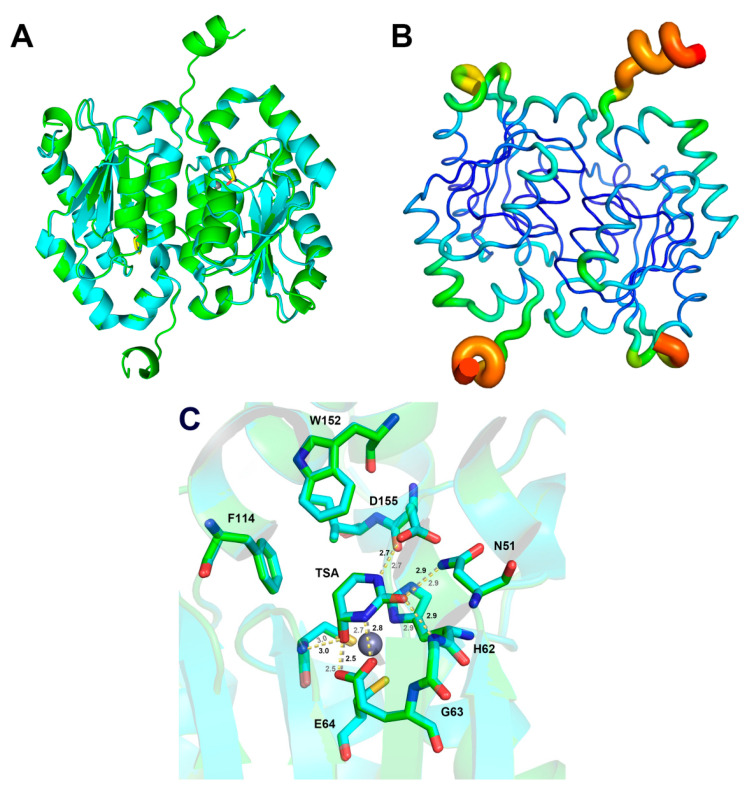
Overlay of the yCD-RQ-1/8SAH TSA complex structure (green, PDB ID: 8I3P) with the wt-yCD TSA complex (cyan, PDB ID: 1P6O): (**A**) overview of yCD-RQ-1/8SAH complex; (**B**) yCD-RQ-1/8SAH colored according to the B-factor with blue for the lowest B-factor and the red for the highest; (**C**) overlay of the two active sites. The hydrogen bond interactions in the active site of yCD-RQ-1/8SAH are shown by a yellow dashed line and those in wt-yCD are shown in grey. The distances between TSA and surrounding residues are labeled with color grey for yCD-RQ-1/8SAH and black for wt-yCD.

**Figure 7 ijms-24-06592-f007:**
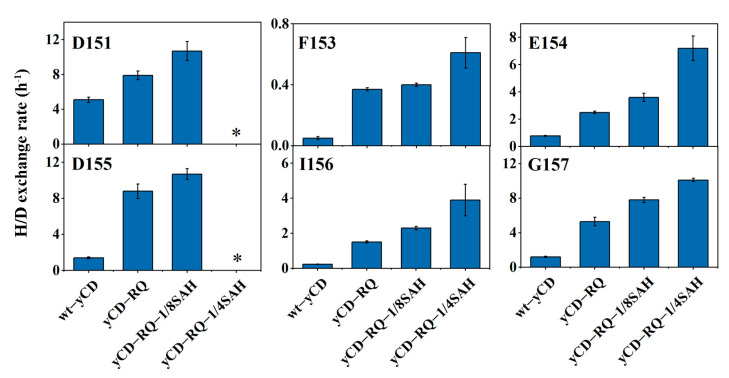
Backbone amide H/D exchange rates of residues in the C-terminal helix of wt-yCD, yCD-RQ, yCD-RQ-1/8SAH, and yCD-RQ-1/4SAH complexed with the TSA. * The exchange rate is faster than the detection limit (~15 h^−1^). The error bars are standard deviation from duplicate experiments.

## Data Availability

The original contributions presented in the study are included in the article. Further inquiries can be directed to the corresponding author.

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
