# Peer review of "Engineering the Active Site Lid Dynamics to Improve the Catalytic Efficiency of Yeast Cytosine Deaminase"

_ijms, 2023, doi:10.3390/ijms24076592_

Round 1

Reviewer 1 Report

It is an interesting article dealing with improvement of catalytic efficiency of yCD on 5-FC to 5-FU via engineering the active site lid opening/close dynamics.  Authors present their findings in a well-organized manner.  Still some questions should be clarified.  (a) What is the role of the linker RQ? (b) Without the linker RQ, can SAH still improve the catalytic efficiency?  (c)From the reported result, relative activity reaches a maximum at yCD-RQ-1/2SAH.  Is this really a maximum?  or after 34+5 residues, the relative activity remains the same?

Author Response

(a) What is the role of the linker RQ?

Response: Attaching SAH to yCD directly may cause potential steric clashes because SAH is rigid. We expect that the linker RQ provides some room for yCD to accommodate SAH. Two sentences have been added on p.9, starting with “The RQ linker was…”.

(b) Without the linker RQ, can SAH still improve the catalytic efficiency? 

Response: We have measured the activity of yCD-SAH (without RQ) which is ~ 2.3 fold of the WT-yCD and smaller than that of yCD-RQ-SAH (Figures S1 and 1). It is not that bad, though. One sentence has been added on p.9, starting with “Although, attaching SAH to…”.

(c)From the reported result, relative activity reaches a maximum at yCD-RQ-1/2SAH.  Is this really a maximum?  or after 34+5 residues, the relative activity remains the same?

Response: The variant yCD-RQ-3/4SAH was prepared. It activity is ~90% of yCD-RQ-1/2SAH, suggests that the yCD variants have the maximum activity at yCD-RQ-1/2SAH. The activity data of yCD-RQ-3/4SAH has been added in Figure 1C.

Reviewer 2 Report

The authors have engineered an extended alpha-helix (SAH) to the C-terminal end of the yCD enzyme to increase its overall 5FC-5FU catalysis rate. The extra SAH fragment enhanced the yCD C-terminal lid dynamics, which favors the release of the catalytic product, enhancing the enzyme turnover. Product release is the rate-limiting step in the WT yCD enzyme. Therefore, the authors have acted on the yCD dynamics rather than on the enzyme catalytic site to improve the yCD enzymatic capacity. The presented results directly affect cancer treatment because the resulting yCD variants improve on previous variants already tested in clinical trials. The authors have optimized the yCD-SAH linker by brute force screening mutagenesis monitored by 5-FC to 5-FU reaction. Also, the optimal length of the SAH helix was described by NMR and kinetic measurements. Authors found 1/4SAH and 1/2SAH provide the best results. I found the article well-structured and clearly explained. The experimental workflow is well-defined, for which I found no critical flaws. The Material and Methods section is clearly explained, for I thank the authors. I congratulate the authors for the quality and straightforwardness of the research and the manuscript presented.

Some minor considerations:

1) The authors have not addressed possible variations of the expression levels of the different yCD variants. While I expect these to be minimal, it is worth addressing this point in the paper to clarify if the different variants pose any toxicity, at least in E. coli.

2) To further foster open data practices, I kindly ask the authors to provide CSV files with the raw data for all plots presented in the article. These can accompany the Supplementary Material.

Small typos:

- Line 268: "its variants"

- Line 401: a word is missing here: "The collected data was and processed with Aquarium"

Author Response

1) The authors have not addressed possible variations of the expression levels of the different yCD variants. While I expect these to be minimal, it is worth addressing this point in the paper to clarify if the different variants pose any toxicity, at least in E. coli.

Response: We did not observer any dramatic change in yCD protein expression level among different variants. An increase of express (by less than 1-fold compared to the WT) was noticed for 15N labeled yCD variants with longer helix tails probably due to better solubility. Toxicity seems not a problem. A brief discussion has been added on p.10, starting with “As the yCD activity increases….”.

2) To further foster open data practices, I kindly ask the authors to provide CSV files with the raw data for all plots presented in the article. These can accompany the Supplementary Material.

Response: The CSV files for all the data plots have been provided.

Small typos:

- Line 268: "its variants"

Response: Corrected.

- Line 401: a word is missing here: "The collected data was and processed with Aquarium"

Response: Corrected.